# Unveiling the intercompartmental signaling axis: Mitochondrial to ER Stress Response (MERSR) and its impact on proteostasis

Jeson J. Li[1,2], Nan Xin[1], Chunxia Yang[1], Bo G. Kim[1], Larissa A. Tavizon[2], Ruth Hong[3], Jina Park[4], Travis I. Moore[1,2], Rebecca George Tharyan[5], Adam Antebi[5,6], Hyun-Eui Kim[1,2]*

1 Department of Integrative Biology and Pharmacology, McGovern Medical School, University of Texas Health Science Center at Houston, Houston, Texas, United States of America, 2 The University of Texas MD Anderson Cancer Center UTHealth Houston Graduate School of Biomedical Sciences, Houston, Texas, United States of America, 3 George R. Brown School of Engineering, Rice University, Houston, Texas, United States of America, 4 Wiess School of Natural Sciences, Rice University, Houston, Texas, United States of America, 5 Max Planck Institute for Biology of Ageing, Cologne, Germany, 6 Cologne Excellence Cluster on Cellular Stress Responses in Aging-Associated Diseases (CECAD), University of Cologne, Cologne, Germany

☙ These authors contributed equally
* hyun-eui.kim@uth.tmc.edu

## Abstract

Maintaining protein homeostasis is essential for cellular health. Our previous research uncovered a cross-compartmental Mitochondrial to Cytosolic Stress Response, activated by the perturbation of mitochondrial proteostasis, which ultimately results in the improvement of proteostasis in the cytosol. Here, we found that this signaling axis also influences the unfolded protein response of the endoplasmic reticulum (UPR$^{ER}$), suggesting the presence of a Mitochondria to ER Stress Response (MERSR). During MERSR, the IRE1 branch of UPR$^{ER}$ is inhibited, introducing a previously unknown regulatory component of MCSR. Moreover, proteostasis is enhanced through the upregulation of the PERK-eIF2α signaling pathway, increasing phosphorylation of eIF2α and improving the ER's ability to handle proteostasis. MERSR activation in both polyglutamine and amyloid-beta peptide-expressing *C. elegans* disease models also led to improvement in both aggregate burden and overall disease outcome. These findings shed light on the coordination between the mitochondria and the ER in maintaining cellular proteostasis and provide further evidence for the importance of intercompartmental signaling.

## Author summary

Cells are constantly at work to maintain proteins and their functionalities. Without proper maintenance systems in place, proteins can become damaged and

**Data availability statement:** All relevant data are within the manuscript and its Supporting Information files.

**Funding:** This research was supported by the University of Texas Health Science Center at Houston (37516-12002 to HK), the Rising STARs program of Texas Health Science Center at Houston (26532 to HK), and the National Heart, Lung, and Blood Institute K01 (NHLBI K01HL143111 to TIM). The funders had no role in study design, data collection and analysis, decision to publish, or manuscript preparation. JJL, NX, CY, and BGK's salaries were supported by the University of Texas Health Science Center at Houston (37516-12002 to HK).

**Competing interests:** The authors have declared that no competing interests exist.

aggregate into adverse clumps–a characteristic commonly found in Alzheimer's and Huntington's disease. Our earlier research discovered that in such cases of stress, the mitochondria send signals to the rest of the cell to help manage protein buildup. These signals also reach a compartment called the endoplasmic reticulum, which assists in folding and processing proteins. The disruption of the mitochondria in *C. elegans,* a model organism that shares many commonalities with the human genome, resulted in a similar protective response; signals from the mitochondria resulted in reduced pressure on protein-folding systems to help manage proteotoxicity within the cell. Our findings reveal a new form of communication between the mitochondria and the endoplasmic reticulum that helps maintain protein balance. Understanding this process may one day lead to new ways to treat diseases that are caused by harmful protein buildup.

## Introduction

A cornerstone of maintaining physiological health is the ability of our cells to maintain protein homeostasis (proteostasis) during challenging times. Our cells have developed unique defense mechanisms to detect and resolve proteotoxic stress through the activation of complex signaling pathways. These pathways mediate the upregulation of chaperone proteins to aid in the proper folding of the misfolded proteins in addition to reducing protein load by suppressing global protein translation [1]. Furthermore, proteolytic systems, such as the ubiquitin-proteasome system and autophagy, play a crucial role in eliminating damaged or misfolded proteins, preventing their accumulation and potential toxicity [1,2]. In particular, the endoplasmic reticulum-associated degradation (ERAD) pathway is essential for recognizing and targeting misfolded proteins within the ER for degradation, thereby contributing significantly to proteostasis [3,4]. Throughout evolution, our cells have evolved to compartmentalize, yielding functionally unique spaces to enhance the efficiency of biological functions [5]. With this also came the rise of compartmental stress responses within the ER, mitochondria, and cytosol. For quite some time, these cellular stress responses have been thought to be stand-alone systems, each encompassing its own unique set of regulatory elements and chaperone proteins. However, recent findings suggest the presence of a regulatory network that mediates an intercompartmental signaling axis.

Previously, we discovered that perturbation of the mitochondrial UPR (UPR$^{mt}$) through RNAi targeting mitochondrial HSP70 (mtHSP70, HSP-6) in *C. elegans* resulted in an increase in the cytosolic heat shock response [6]. This mitochondrial to cytosolic stress response (MCSR) improved cytosolic proteostasis in both *C. elegans* and human cell lines. Further evidence suggested that MCSR is regulated through the alteration of lipid metabolism, specifically ceramide and cardiolipin [6,7]. Although this was the first evidence of interaction between the mitochondria and the cytosol mediated through *mtHSP70*, previous interactions between the mitochondria and ER have been described. Rizzuto *et al*. first characterized the importance of mitochondria and ER contact sites (MERCS) in the regulation of calcium signaling [8]. MERCS has

since been well studied and its role in multiple biological functions including $Ca^{2+}$ signaling [9], lipid transport [10], mitochondria dynamics [11], and stress response [12] has been characterized. MERCS has also been implicated in an array of diseases including neurodegenerative disease [13] and cancer [9,14] and is currently being explored as a possible therapeutic target [15].

The ER is the major site of secretory protein synthesis, and the proper function of the UPR[ER] is crucial to maintaining proper cellular function [16]. While *hsp-6 (C. elegans* homolog of *mtHSP70)* knockdown is able to induce MCSR and offers insight into the intercompartmental signaling axis between the mitochondria and cytosol, the question remains whether this axis exerts a dualistic influence over the ER. Indeed, our data suggests that the knockdown of *hsp-6* modulates UPR[ER] by potentiating UPR[ER]-specific signaling pathways. Using animals harboring a transcriptional reporter of UPR[ER] (*hsp-4*p::GFP), we observed a decrease in GFP fluorescence when treating the worms with tunicamycin followed by *hsp-6* knockdown. Recent evidence suggests a regulatory role of mitochondrial proteins on the UPR[ER] and Integrated Stress Response (ISR) [17–21]; however, the influence of the mitochondria on the UPR[ER] is currently a field that has not been well studied. Hence, more extensive work is needed to fully characterize this signaling axis.

We have previously shown that *hsp-6* knockdown reduces polyglutamine aggregates in a *C. elegans* strain that expresses polyglutamine repeats within body wall muscles [22]. We hypothesize that this decrease in aggregate abundance occurs partially through the activation of the UPR[ER] signaling pathways. The activation of the UPR[ER] signaling pathway increases the protein folding capacity of the ER, which compensates for the increase in protein folding stress and ultimately helps to achieve ER stress resolution. When we perturbed the UPR[ER] signaling pathway during *hsp-6* knockdown, we surprisingly found that the IRE1 branch of UPR[ER] was inhibited. However, we found a significant increase in PERK-dependent eIF2alpha (eIF2α) phosphorylation, suggesting a decrease in global protein translation levels. Concomitantly, ER protein secretion capacity was also shown to be downregulated, suggesting a functional inhibition of the ER to further prevent the increase in misfolded proteins. Taken together, we infer that Mitochondria to ER Stress Response (MERSR) enhances proteostasis by dampening the IRE1 branch of the UPR[ER] signaling pathway, reducing protein folding stress through the suppression of global protein translation and ER protein secretion. This ultimately increases the ER's capacity to manage proteostasis challenges.

## Results

### Tunicamycin-induced ER stress is suppressed through the activation of MERSR

To determine the effect of *hsp-6* knockdown on UPR[ER], we utilized a *C. elegans* UPR[ER] reporter strain where a GFP tag has been fused to the promoter of a UPR[ER] chaperone (SJ4005, *hsp-4*p::GFP). Tunicamycin treatment of the worms induced ER stress, resulting in GFP fluorescence that can be measured and quantified using fluorescent microscopy. ER stress reporter worms were treated with tunicamycin during day 1 of adulthood followed by transfer onto RNAi plates containing *hsp-6* targeting RNAi sequences. Loss of *hsp-6* resulted in a reduction of GFP fluorescence post-tunicamycin treatment when compared to empty vector control (Fig 1A). To determine if the UPR[ER] suppression by *hsp-6* knockdown was limited to tunicamycin, we targeted 2 additional distinct UPR[ER] activators. 1) NFYB-1 is a histone-like transcription factor that regulates mitochondrial function. Interestingly, it was found that NFYB-1 knockout exhibited cross-compartmental regulation and activation of UPR[ER] [7]. 2) VCP is an AAATPase that among many other things, regulates ER-associated degradation (ERAD) [21]. The knockout of VCP is known to induce UPR[ER] due to increased proteotoxic stress as a result of the increase in misfolded proteins [23]. Ultimately, *hsp-6* knockdown was able to reduce the UPR[ER] induced by loss of either NFYB-1 (Fig 1B) or CDC-48, a *C. elegans* homolog of VCP (Fig 1C), suggesting that downregulation of *hsp-6* reflects an intrinsic mechanism that regulates an intercompartmental UPR signaling axis during times of stress. Interestingly, when we target transcription factors involved in the UPR[mt] signaling pathway, such as *dve-1*, a similar reduction in UPR[ER] reporter fluorescence level is observed (S1A Fig), suggesting that the reduced UPR[ER] activation is a direct consequence of alterations that originate within the mitochondria. Mitochondrial stress activated through *mrps-5*, *spg-7* or *cco-1*

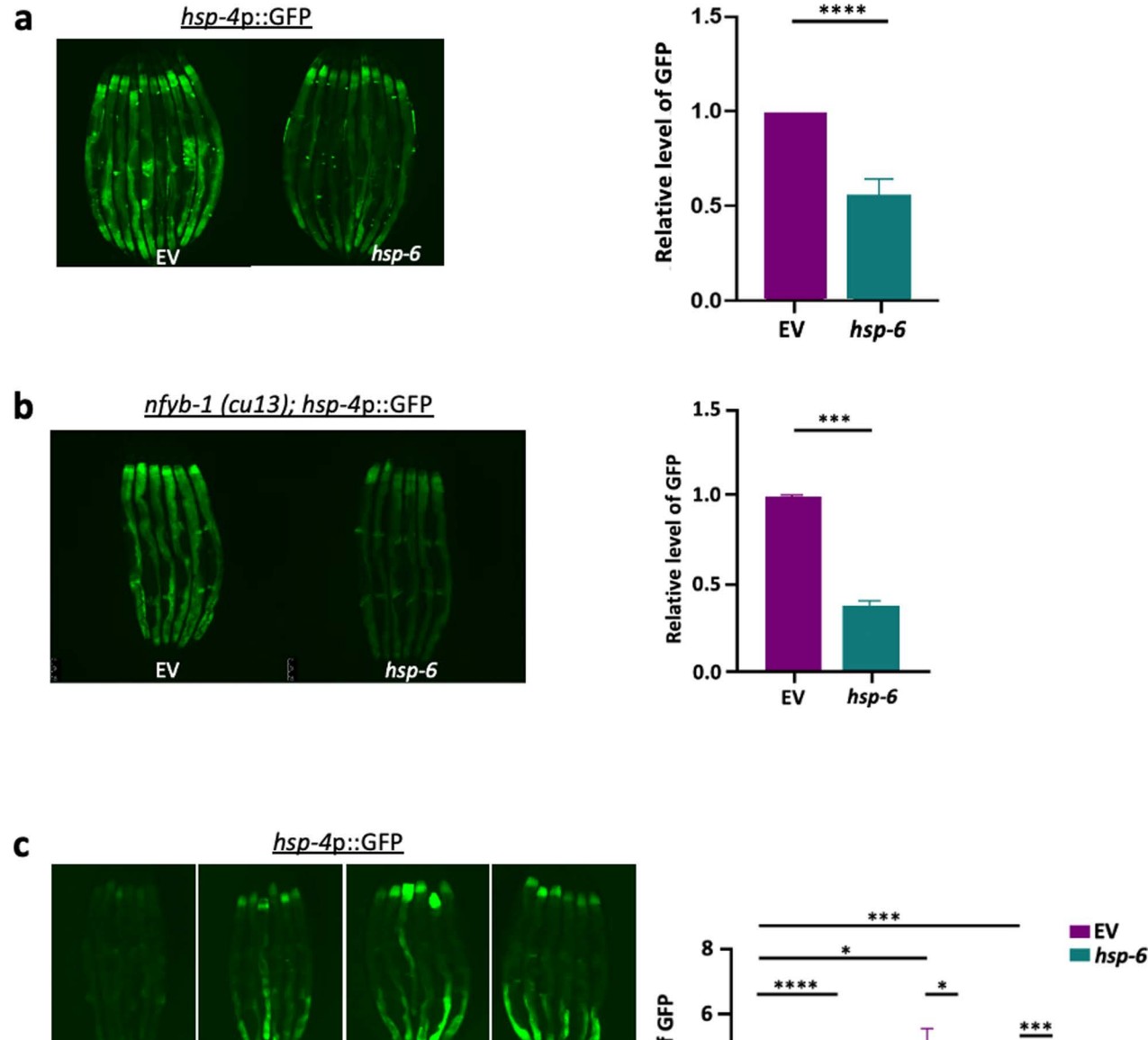

**Fig 1. Inhibition of *hsp-6* results in the reduction of ER stress response.** *hsp-4*p::GFP, the UPR[ER] reporter animals were transferred to the RNAi-containing plates L4 adulthood (55 hours post bleaching). On day 1 adulthood (72 hours post bleaching), animals were treated with **a)** tunicamycin for 4 hours in order to induce ER stress. Following tunicamycin treatment, the worms were transferred onto RNAi-containing plates and grown at 20°C until day 3 of adulthood and were imaged to assess ER stress levels. Other forms of ER stress, including **b)** *nfyb-1 (cu13)* mutant and c) VCP (*cdc-48.1* and *cdc-48.2*) knockdown, were tested in the same manner to further validate that our observation was not tunicamycin specific but rather a conserved physiological mechanism. When multiple RNAi treatments were needed, the same amount of dsRNA-expressing bacteria were mixed. The RNAi target

genes are as indicated, EV: empty vector control. The graphs show the mean +/-SD of the animals with representative images of animals, n>=6 (three biological repeats). Each RNAi-treated cohort was compared to the EV control to compare GFP induction.

knockdown was unable to reduce UPR$^{ER}$ signaling during tunicamycin treatment (S1B Fig) or in *nfyb-1* mutants (S1C Fig), suggesting that *hsp-6* knockdown triggers a distinct mitochondrial stress that elicits a signaling cascade between the mitochondria and the ER. Interestingly, when both the ER and mitochondria were perturbed by treating animals with thapsigargin, which depletes ER calcium stores and increases cytosolic calcium levels, leading to mitochondrial calcium overload [24,25], *hsp-6* knockdown further enhanced UPR$^{ER}$ reporter expression (S1D Fig). This suggests that UPR$^{ER}$ is differentially regulated depending on the type of trigger and that calcium signaling between the mitochondria and ER is not modulated by MERSR. Additionally, ER stress induced by tunicamycin or the loss of *nfyb-1* did not elevate HSP-6 levels (S2A and S2B Fig), ruling out the possibility of ER-to-mitochondrial stress communication via these ER stressors.

Collectively, our data suggest that *hsp-6* knockdown not only activates MCSR, but also induces a cross-compartmental signaling axis, triggering a mitochondrial-to-ER stress response (MERSR).

## Lipid metabolism mediates the ER suppression by *hsp-6* RNAi

We previously observed that MCSR is mediated through altered lipid metabolism, in particular, the downregulation of ceramide and the upregulation of cardiolipins [22]. In *C. elegans*, *de novo* ceramide synthesis depends on the function of three separate ceramide synthase enzymes HYL-1, HYL-2 and LAGR-1, and the knockdown of these enzymes can alter the global sphingolipid landscape of the animals [26]. To investigate the possibility that sphingolipid metabolism also regulates MERSR, we targeted several critical enzymes in the *de novo* sphingolipid synthesis pathway in addition to cardiolipin synthase (*crls-1*) through RNAi. By inducing UPR$^{ER}$ in ER stress reporter worms with tunicamycin treatment followed by the perturbation of key sphingolipid synthesis enzymes through RNAi, we found that the knockdown of ceramide synthases led to a decrease in UPR$^{ER}$ reporter intensity, similar to that of *hsp-6* RNAi (Fig 2A). In addition, the *hyl*-1 deletion mutant also reduced UPR$^{ER}$ activation (S2C Fig), further supporting that ceramide synthase activity is critical for suppressing UPR$^{ER}$. However, other enzymatic targets within the *de novo* synthesis pathway did not yield such results (Fig 2B). Conversely, cardiolipin synthase (*crls-1*) knockdown negated the inhibition of UPR$^{ER}$ through *hsp-6* knockdown (Fig 2C, the efficiency of *hsp-6* RNAi in the double knockdown is shown in S2D Fig). These results suggest that not only do ceramide and cardiolipin mediate the stress signaling response between the mitochondria and the cytosol, but they also regulate the UPR$^{ER}$ response initiated through the knockdown of *hsp-6.* The decrease in UPR$^{ER}$ reporter signaling due to the knockdown of ceramide synthase could be the result of increased UPR$^{ER}$ threshold caused by changes in ceramide levels, enhancing the capability of the cells to combat protein stress and, therefore, reducing UPR$^{ER}$ [27]. Furthermore, *nfyb-1* deleted mutant animals constitutively activate higher extent of UPR$^{ER}$ (S1C Fig), which could be reduced through the activation of MERSR (Fig 1B). Interestingly, the lipid profiles of these *nfyb-1* mutants exhibit a high level of ceramide and low levels of cardiolipin compared to their wild-type counterparts [7]. The reduction in UPR$^{ER}$ observed in these mutant animals through MERSR activation could be the result of reversing the ceramide and cardiolipin content, reducing ceramide while increasing cardiolipins to reduce lipid signals activating UPR$^{ER}$.

## MERSR is mediated through the UPR$^{ER}$ signaling pathway

UPR$^{ER}$ signaling is mediated through three separate stress sensory pathways that work together to increase the capacity of the ER to combat proteotoxic stress [1]. During protein folding stress, membrane IRE1 oligomerizes, initiating splicing of XBP-1 mRNA to the active transcription factor XBP-1s, resulting in the upregulation of various genes, including major chaperone proteins in order to aid in the proper folding and recycling of misfolded proteins [28,29]. In addition to increasing chaperone levels, another major compensatory mechanism of the ER is the PERK-eIF2α signaling pathway,

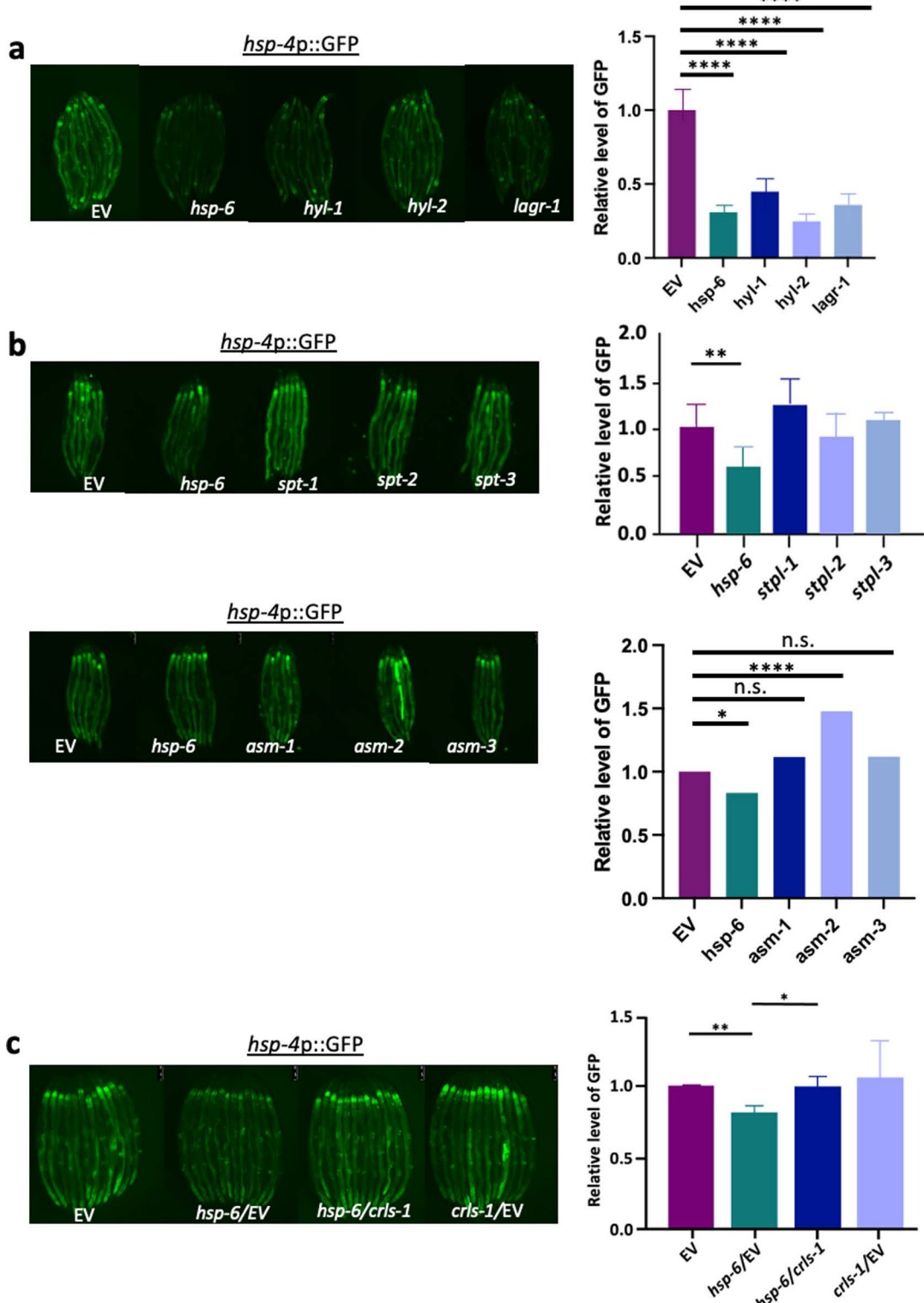

**Fig 2. The reduction of ER stress by *hsp-6* RNAi is partially regulated through lipid metabolism.** **a)** *hsp-4*p::GFP, the UPR<sup>ER</sup> reporter animals were used to determine the effects of *hyl-1*, *hyl-2* and *lagr-1* on UPR<sup>ER</sup> induction as described in Fig 1. **b)** Other sphingolipid enzymes along the *de novo* synthesis pathway and the salvage pathways were also tested in the same manner. **c)** Knockdown of cardiolipin synthetase *crls-1* positively regulated

induction of UPR<sup>ER</sup>. When multiple RNAi treatments were needed, the same amount of dsRNA-expressing bacteria were mixed. The RNAi target genes are as indicated, EV: empty vector control. The graphs show the mean+/-SD of the animals with representative images of animals, n>=6 (three biological repeats). Each RNAi-treated cohort was compared to the EV control to compare GFP induction.

which attenuates global protein translation levels and reduces the total protein load of the ER [30]. To determine the effects of MERSR on UPR$^{ER}$ signaling pathways, we perturbed the UPR$^{ER}$ signaling pathway during *hsp-6* knockdown and found that the double knockdown of *hsp-6* with *ire-1* or *xbp-1* resulted in the loss of MERSR activation (Fig 3A). Proper UPR$^{ER}$ signaling is reliant on the dimer/oligomerization of IRE1, which initiates splicing of *xbp-1* mRNA. To investigate the impact of *hsp-6* knockdown on ectopic UPR$^{ER}$ activation at two different points within the same pathway, we utilized 1) overexpression of IRE1 lacking the luminal domain, which activates UPR$^{ER}$ in the absence of protein folding stress but still requires proper dimer/oligomerization and 2) overexpression of *xbp-1*s (active form of *xbp-1*), allowing UPR$^{ER}$ activation while bypassing IRE1 dimer/oligomerization (Fig 3B). Overexpression of IRE1 lacking the luminal domain (*ire-1a* (344–967aa)) in the intestine induced UPR$^{ER}$ reporter [31]. This induction was moderately reduced upon *hsp-6* knockdown, whereas *hsp-6* knockdown had no impact on *xbp-1*s overexpression-induced UPR$^{ER}$ (Fig 3C and 3D). Based on these data, we conclude that mitochondrial stress response induced by *hsp-6* knockdown during adulthood interferes with UPR$^{ER}$ at the upstream IRE1 level, and once *xbp1-s* is produced, mitochondria can no longer suppress the downstream UPR$^{ER}$ activation.

During times of protein folding stress, the activation of the UPR$^{ER}$ have also been shown to decrease global protein load through the phosphorylation of PERK-eIF2α signaling cascade [32]. Thus, we examined the levels of eIF2α phosphorylation during *hsp-6* knockdown. Along with PERK, we also targeted *gcn-2*, a PERK-independent eIF2α kinase activated primarily through amino acid starvation [33] and reactive oxygen species produced during mitochondrial stress [18]. Our data show that *hsp-6* knockdown enhances eIF2α phosphorylation levels (Fig 3E). However, *gcn-2* knockdown had no effect on either the induction or the suppression of UPR$^{ER}$ upon tunicamycin treatment (S3A Fig). This suggests that MERSR increases eIF2α phosphorylation in a PERK-dependent manner. Furthermore, when we performed a double knockdown of *hsp-6* and *pek-1*, the inhibition of UPR$^{ER}$ was no longer observed (S3A Fig). This indicates that MERSR is PERK-dependent. Given that PERK can also repress XBP1 targets [34], it is possible that both the IRE1/XBP1 and PERK/eIF2α branches contribute to proteostasis, either directly or indirectly. To further investigate this, we knocked down UPR$^{ER}$ components in the animals expressing amyloid-β (Aβ$_{1-42}$) or polyglutamine (S3D Fig). We observed that *xbp-1* knockdown exacerbates proteotoxicity in both models, suggesting that IRE1/XBP1 pathway is required for proteotoxicity resolution when MERSR is not active. Thus, MERSR may enhance eIF2α phosphorylation to increase the threshold for IRE1 activation, shifting the stress response balance. Altogether, our data suggest that *hsp-6* knockdown enhances ER proteostasis by increasing eIF2α phosphorylation via PERK signaling, which in turn reduces global protein translation to mitigate ER stress.

Given that MERSR reduces global protein translation through the mediation of UPR$^{ER}$ signaling, we further assessed the function of the UPR$^{ER}$ by investigating its secretory capacity regulated by the IRE1 signaling pathway. In *C. elegans*, DAF-28 is one of 40 insulin-like peptides [35]. DAF28::GFP fusion protein is processed within the ER and secreted to the body cavities of the worms [36], eventually absorbed by the coelomocytes, phagocytic and macrophage-like cells that engulf surrounding materials for degradation [37]. GFP signal within the coelomocytes can be used as a readout to determine the ER secretion capacity of the cells [38]. We observed that the knockdown of *hsp-6* greatly reduced the GFP fluorescence within the coelomocytes (Fig 3F) to a level similar to that of ER dysfunction induced through the knockdown of *ire-1* and *xbp-1*. Western blot analysis of both transcriptional and translational *daf-28* expression reporter worms further confirmed that the reduction observed in the coelomocytes during *hsp-6* RNAi was the result of decreased secretion rather than changes in the innate expression of *daf-28* (S3B Fig). Consistent with earlier results, *cco-1* RNAi did not impact DAF-28 secretion (S3B Fig). Additionally, UPR$^{ER}$ induced by the protein trafficking inhibitor, Brefeldin A, was

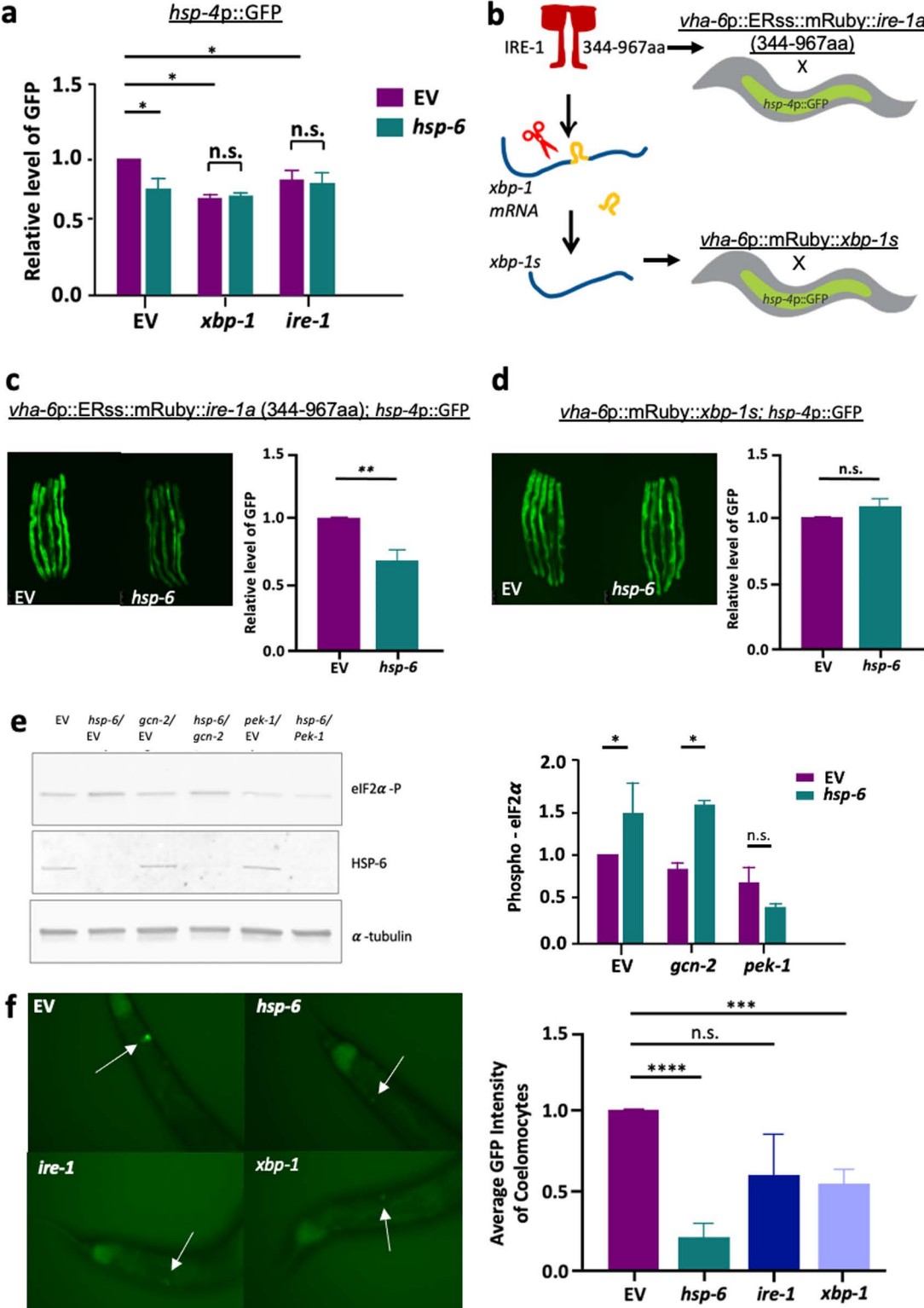

**Fig 3. The reduction of ER stress by *hsp-6* RNAi is the result of IRE-1 and PERK modulation. a)** Tunicamycin-treated animals induced UPR^ER in an *xbp-1* and *ire-1* dependent manner. **b)** *hsp-4*p::GFP, the UPR^ER reporter strain was crossed with animals that constitutively activate UPR^ER through the IRE-1-XBP-1 pathway (drawn by hand, Jeson J Li) using Microsoft PowerPoint and its icons). Animals expressing constitutively active IRE-1a

(344-967aa) that are lacking part of the luminal domain and animals expressing *xbp-1s* both activate UPR[ER] (c and d). **c and d)** *hsp-6* RNAi reduced UPR[ER] activation only in the animals expressing constitutively active IRE-1a (c) but not in the animals expressing the *xbp-1* spliced form*, xbp-1s*. MCSR regulates UPR[ER] at the IRE-1 level on the ER membrane. Animals were transferred to *hsp-6* RNAi plates or the empty vector control plates on late L4/ early adult and were imaged on day 3 (72 hours of RNAi treatment). **e)** Wild-type N2 animals were treated with EV, *gcn-2,* or *pek-1* RNAi from hatch to the late L4 stage, before transferring to double RNAi treatment or to continue single RNAi treatment. Two individual double RNAi treatment groups were set up utilizing a 1:1 ratio of EV (purple) or *hsp-6* (teal) combined with *gcn-2* or *pek-1* for overnight treatment. *Hsp-6* RNAi increased the eIFα phosphorylation in a *pek-1*-dependent manner. **f)** DAF-28::GFP transgenic animals were treated with indicated RNAi, or the empty vector control from day 1 of adulthood. Animals were imaged for coelomocyte GFP content on day 4. The graph shows mean+/-SD of GFP intensity normalized to empty vector control, n>=6 with three biological repeats.

suppressed by *hsp-6* knockdown, likely due to MERSR-mediated reduction of protein translation and secretion (S3C Fig). These data suggest that in addition to decreasing global translation through the upregulation of PERK-eIF2α signaling, *hsp-6* knockdown also inhibits ER secretory functions. This reduction in ER protein secretion could serve as a compensatory mechanism during times of stress to further slowdown the proteotoxic flux.

## MERSR and ceramide reduction can benefit both ER and cytosolic stress-related neurodegenerative diseases

Dysfunctional ER stress is a hallmark of many neurodegenerative diseases, including Alzheimer's and Parkinson's disease [39]. While mild UPR[ER] activation can initially mitigate amyloid-β (Aβ$_{1-42}$) accumulation, chronic ER stress exacerbates neurodegeneration [2,40,41]. Our findings suggest that MERSR and ceramide reduction alleviate both ER and cytosolic stress, potentially benefiting neurodegenerative disease models by improving proteostasis and reducing apoptotic signaling.

Therefore, we explored the impact of MERSR on Aβ$_{1-42}$ aggregate burden in *C. elegans*. We hypothesized that the pro-proteostasis phenotype observed with MERSR could decrease the Aβ$_{1-42}$ burden, leading to improvements in the overall disease state. To evaluate the physiological effects of the Aβ$_{1-42}$ burden, neuronal RNAi-enabled worms expressing pan-neuronal Aβ$_{1-42}$ were put under the effects of mild heat. At 25°C, the increased temperature accelerates the aggregate formation process, leading to rapid paralysis of the worms. The neuronal knockdown of *hsp-6* was able to delay the rate of paralysis, suggesting a decrease in Aβ$_{1-42}$ aggregate burden (Fig 4A). Consistently, animals expressing Aβ$_{1-42}$ within the body wall muscle showed moderate improvement of paralysis at 25°C (S4B Fig). Similarly, the knockdown of ceramide synthases (*hyl-1* and *hyl-2)* also improved the paralysis of neuronal Aβ$_{1-42}$ -expressing animals. In addition, *hyl-1* knockdown also improved the lifespan of those animals (Fig 4C), providing further evidence that lipid metabolism plays an integral role in the regulation of cross-compartmental stress response.

*hsp-6* knockdown moderately extends the lifespan of wild-type animals in the absence of ER stress (S4C Fig). On the other hand, *hsp-6* knockdown did not extend the lifespan of the UPR[ER]-activated *nfyb-1* mutant animals (S4C Fig), suggesting that MERSR may help proteostasis recovery with ongoing ER stress but may not be able to extend the lifespan of the stressed animals.

Huntington's Disease is characterized by the expansion of the polyglutamine tracks in mutant Huntingtin (htt) proteins, resulting in neurotoxic htt aggregate formation [43]. These aggregates inhibits the ER-associated degradation (ERAD) pathway, increasing proteotoxic stress and activating UPR[ER] [39]. To determine whether MERSR enhances proteostasis in polyglutamine-expressing animals, we examined animals expressing neuronal polyglutamine (Q40::YFP) repeats. Activation of MERSR through *hsp-6* knockdown, or *hyl-1* knockdown, significantly improved motility, as evidenced by an increase in body bends (Fig 4B). Similarly, the expression of polyglutamine (Q35::YFP) in the body wall muscle showed reduced polyglutamine aggregation upon MERSR activation (Fig 4D and 4E). Interestingly, this reduction in polyglutamine aggregates required VCP (CDC-48 in *C. elegans*), a key component of ERAD, and the ubiquitin-proteasome system (UPS) (Fig 4D and 4E). Therefore, we examined the colocalization of VCP with densely packed (cyan) and loosely packed (yellow) polyglutamine aggregates by the staining intensity. Immunostaining of VCP and polyglutamine aggregates in

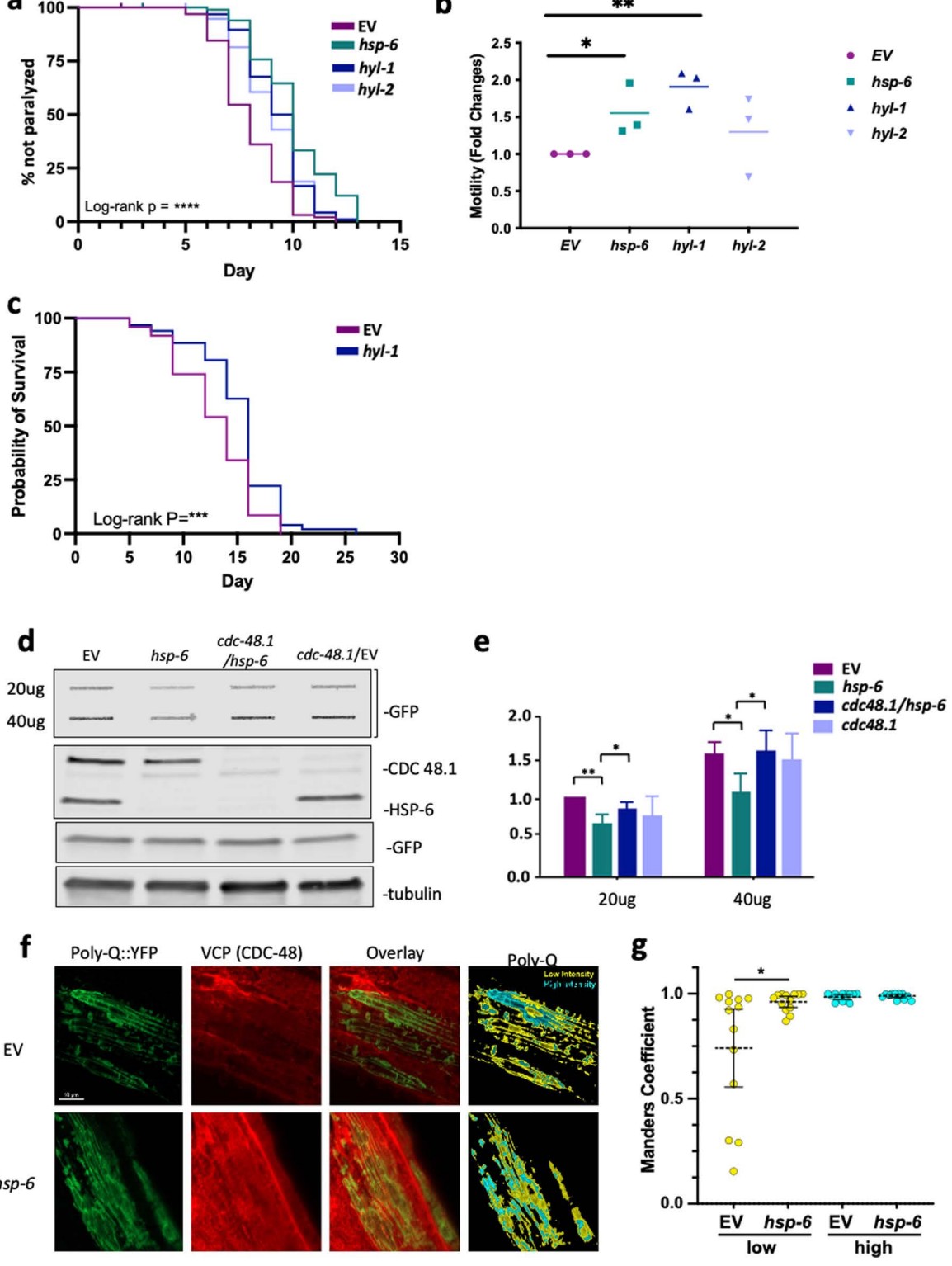

**Fig 4. The knockdown of *hsp-6* reduces protein aggregation in poly-glutamine disease model worms. a)** Paralysis of pan-neuronal Aβ expressing animals crossed with neuronal RNAi-enabled line (GRU102;TU3401) at 25°C. Treatment of RNAi against *hsp-6, hyl-1, hyl-2 and* EV were used to determine improvements on paralysis during heat shock. Log-rank P value was obtained comparing RNAi to empty vector control: P<0.0001 for

all *hsp-6*, *hyl-1*, and *hyl-2*. **b)** Animals expressing polyglutamine aggregates (Q40::YFP) within the neurons (AM101) were crossed with a neuronal RNAi-sensitive worm line (TU3401). Motility was determined using a body bending assay that measures the number of body bends per 30 seconds. The relative motility was plotted by normalizing with empty vector control, and the rate was compared to that of the empty vector control (Three biological repeats with n>12, mean +/- SD). **c)** Lifespan of the animals expressing Aβ in neurons with neuronal-sensitive RNAi treatment as shown in a), GRU102; TU3401. P<0.001. * TU3401 strain is known to exhibit RNAi effects in non-neuronal tissues at later ages as it becomes "leaky"; a tighter controlled line has been developed recently [42]. Nonetheless, this strain will have neuronal RNAi knockdown of the target genes, however, it is possible to have RNAi effects in non-neuronal tissues at later ages. **d)** The accumulation of poly-glutamine aggregates is reduced upon MERSR. Animals expressing polyglutamine aggregates (Q35::YFP) in the body wall muscle (AM140) were treated with indicated RNAi or the empty vector control from day 1 adulthood. Animals were collected on day 5. Protein aggregates were measured by applying protein lysate onto a cellulose acetate membrane through a vacuum slot blotter. The membrane was blotted with GFP antibody to detect the Q35::YFP aggregates. 20ug of total protein lysate from each RNAi treatment was applied to SDS-PAGE followed by Western blot of alpha-tubulin, GFP, VCP, and HSP-6 antibody (lower panels). Total protein and total Q35::YFP protein levels were both at equivalent levels across all specimens. The VCP blot and HSP-6 blot showed that the RNAi knockdown of the targeted protein worked efficiently. **e)** Graph shows mean+-SD of filter trapped Q35::YFP from three biological repeats. **f)** Representative confocal image of polyQ::YFP (green) and VCP (CDC48, red) localization in the body wall muscle of AM140. Segmentation of polyQ::YFP by emission intensity (low, yellow; high, cyan). **g)** Colocalization of VCP with polyQ as measured by the Meanders correlation in high (yellow in the segmentation image) and low-intensity (cyan in the segmentation image) areas of polyglutamine (Poly-Q::YFP) in the empty vector control and *hsp-6* knockdown worms. Bars show mean ± 95% CI. Two-tailed Mann–Whitney tests show p=0.05 (*). n =13 (EV); n=14 (*hsp-6*).

body wall muscle revealed VCP is predominantly associated with densely packed aggregates compared to loosely packed aggregates ([Fig 4F] and [4G]). However,

in *hsp-6* RNAi-treated animals, there was a notable increase in VCP colocalization with loosely packed aggregates ([Fig 4F] and [4G]). This observation supports the role of MERSR in modulating protein quality control mechanisms.

Finally, we investigated the impact of excessive UPR$^{ER}$ activation on proteostasis by crossing *xbp-1s*-overexpressing animals with polyglutamine-expressing worms and subjecting them to lifespan and motility assays ([S3A Fig]). These worms exhibited a significantly shortened lifespan, and none of the RNAi treatments that typically alleviate proteotoxicity were able to extend it. Notably, *hyl-2* RNAi further exacerbated the lifespan reduction. Consistent with the lifespan data, none of the indicated RNAi treatments improved motility; instead, they further impaired it. Therefore, excessive UPR$^{ER}$, in combination with MERSR, may be even more detrimental when UPR$^{ER}$ is uncontrollable. These results suggest that uncontrolled UPR$^{ER}$ activation does not benefit cellular physiology and instead requires precise regulation by pathways such as MERSR.

These data suggest that MERSR activation and ceramide perturbation through ceramide synthase knockdown can improve the disease outcome of both cytosolic- and ER-associated neurotoxic aggregate worm lines. Taken together, our findings suggest that the increase in mitochondrial stress triggered through *hsp-6* knockout coordinates both cytosolic and ER stress responses, leading to enhanced cellular proteostasis through the upregulation of UPR$^{ER}$ signaling pathways ([S5 Fig]). Activation of MERSR in proteotoxicity models demonstrates its efficacy and inducibility in post-mitotic tissues such as neurons, further emphasizing the importance of this signaling pathway and the need for future research.

## Discussion

Here, we have identified an unexpected role of MCSR in promoting the balance of proteostasis by regulating the UPR$^{ER}$ signaling pathways in *C. elegans*. Our study suggests that the knockdown of *hsp-6* not only coordinates mitochondria to cytosolic stress response [6] but also elicits an intercompartmental signaling axis between the mitochondria and the ER. Mitochondrial stress triggered by the knockdown of *hsp-6* results in decreased UPR$^{ER}$ inducibility by multiple ER stress inducers, including tunicamycin, VCP knockout, and the loss of NFYB-1. The suppression of the IRE1 branch of the UPR$^{ER}$ is mediated through perturbations of sphingolipid metabolism, suggesting the possibility of bioactive lipids serving as mediators along this intercompartmental signaling axis. The proper function of the UPR$^{ER}$ is dependent on several signaling cascades. The PERK-eIF2α signaling cascade results in an increase in the phosphorylation of eIF2α, leading to a reduction in global protein translation levels. We have shown here that reduced UPR$^{ER}$ inducibility is a direct result of increased PERK-dependent eIF2α phosphorylation in combination with decreased ER protein secretion.

Mitochondrial HSP70 (mtHSP70, mortalin, HSPA9, Grp75, HSP-6) is widely known for its beneficiary functions. It was first described as a substrate for protein translocation into the mitochondrial matrix [44]. However, as research on the subject has increased, the role of mortalin has begun to expand. Mortalin reduces the formation of reactive oxygen species and lipid peroxidation, protecting the mitochondria from oxidative stress [45]. In addition, mortalin exhibits protective properties against neurotoxicity in AD [46] and PD [47] through targeting of Aβ and alpha-synuclein, respectively. In light of this, an insightful review was recently published describing the link between mortalin and neurodegenerative disease [48]. Interestingly, in our study, the post-developmental knockdown of *hsp-6* in *C. elegans* conferred a protective phenotype against proteotoxic stress, suggesting a previously undiscovered role of mortalin. In support of our data, Honrath *et al*. also demonstrated a protective phenotype against glutamate-induced oxidative cell death in Grp75 knockout HT22 cells [49]. However, this protection did not extend to drug-induced ER stress. This contradiction could be explained by the utilization of cancer cells in testing for ER stress protection. Cancer cells exhibit high levels of innate ER stress, and treatment with tunicamycin or thapsigargin may exacerbate ER stress and induce apoptotic signaling. In recent cancer research, mortalin has become a target for drug discovery due to its ability to bind P53 and prevent its nuclear localization [50]. However, this colocalization is not observed in non-cancer cells [51], and the protective phenotype induced through *hsp-6* knockdown is still unknown. Despite this, given the significant changes in lipid metabolism, specifically cardiolipin and ceramide, it is likely that *hsp-6* knockdown-induced mitochondrial stress could result in the remodeling of the lipid composition within the mitochondrial membrane. Cardiolipin is a key regulator in the maintenance of mitochondrial membrane integrity and cristae morphology [52]. Proper cardiolipin metabolism is also responsible for mediating proper protein-protein and protein-membrane interactions [53] in addition to stabilizing the respiratory chain complex [54]. Although the mechanism of regulating cardiolipin to ceramide ratio between the mitochondria and the ER during *hsp-6* knockdown remains undetermined, recent evidence revealed that *hsp-6* is inserted into the mitochondrial membrane and possesses an affinity for negatively charged phospholipids such as cardiolipin [55]. Indeed, cardiolipins are externalized during mitochondrial stress and serve as a signal for the induction signaling for mitophagic [56] and apoptotic [57] signaling; however, the exact mechanism of how cardiolipins regulate cross-compartmental signaling pathways will require further research.

Our data also suggests that the suppression of UPR$^{ER}$ occurs upstream of *xbp-1*, possibly at a membrane level where IRE1 resides. One possibility is that *hsp-6* knockdown leads to an alteration in membrane fluidity, compromising the dimerization of the UPR$^{ER}$ sensor IRE1. This is further corroborated by recent findings that the knockdown of ER-resident proteins alters the ER membrane lipid content, increasing mitochondria and ER contact through increasing membrane order [58]. Furthermore, alterations in sphingolipid, including ceramide content within the ER membrane, have been shown to induce UPR$^{ER}$ in the absence of aberrant proteostasis [59], providing further evidence that membrane lipid content can directly influence the UPR$^{ER}$ signaling pathways.

Furthermore, we have also shown that MERSR induction also improved disease states in our Aβ models. Although the Aβ$_{1-42}$ aggregates we observe in our worm models are primarily cytosolic [60], the expression of Aβ$_{1-42}$ requires the cleavage of a synthetic signaling peptide, which occurs at the ER [61,62]. We have shown that MERSR reduces ER protein secretion. Therefore, it would be prudent to suggest that the cleavage of Aβ$_{1-42}$ may be compromised during MERSR induction, which would result in the decrease in disease burden that we observed in our Aβ worm model. Interestingly, ERAD-related chaperone VCP was associated with loosely packed polyglutamine aggregates in the muscle of polyglutamine expressing animals when MERSR was activated. In these animals, MERSR reduced protein aggregates in a VCP-dependent manner, and the knockdown of VCP resulted in the loss of proteostasis improvement. Although during ERAD, VCP is recruited to the surface of the ER by ubiquitin ligases to aid in the clearance of misfolded proteins [63], recent evidence shows strong support for the colocalization of VCP with neurotoxic aggregates [64]. The regulatory role of VCP in protein homeostasis occurs at multiple levels. During the initial process of aggregate formation, VCP regulates the UPS to detect and ultimately degrade misfolded proteins [65]. However, as aggregate burden increases, VCP can regulate macroautophagy and aid in removing excess protein aggregates and damaged organelles to restore homeostasis

[66]. Our study showed that VCP is essential in reducing polyglutamine aggregates during MERSR; Although the precise downstream effects of VCP mobilization remain unclear, VCP plays a well-established role in aggregate handling. We postulate that its recruitment to polyglutamine aggregates triggers degradation via the UPS or macroautophagy. The inability of xbp-1s-overexpressing worms to recover from proteotoxic stress, despite MERSR activation, highlights the necessity of tightly regulated UPR^ER signaling. Unchecked activation of UPR^ER may contribute to cellular dysfunction rather than protection, emphasizing the importance of balanced proteostasis pathways.

The downstream mechanisms of how aggregate clearance is achieved were beyond the scope of this paper. However, this does leave us with an exciting future research question of how mitochondrial stress regulates VCP activity to improve proteotoxic stress. Recent evidence has already shown VCP's involvement in the regulation of mitochondria and ER contact [67,68], further exacerbating the need for elucidation in this subject.

In conclusion, we have demonstrated for the first time, within our knowledge, the capabilities of a mitochondrial chaperone protein to regulate UPR^ER signaling through the perturbation of lipid metabolism. Current understanding of the relationship between the mitochondria and the ER is incomplete, although research on this subject has increased due to advances in biochemical and fluorescent microscopy. We know that mitochondria and the ER form contact sites in order to modulate membrane lipid composition and proper $Ca^{2+}$ homeostasis. The question that remains is how *hsp-6* knockdown impacts mitochondria and ER contact. Although we did not explore this in our study, altered mitochondria and ER contact is seen in several tumor types as well as several neurodegenerative diseases, highlighting the importance of further research into this field.

## Methods

### Strains and culture

AGD2797 (*unc-119*(*ed3*) III; uthSi71[*vha-6*p::mRuby::*xbp-1s*::*unc-54* 3'UTR, cb-*unc-119*(+)]) and AGD3156 (*unc-119*(*ed3*) III; uthSi65[*vha-6*p::ERss::mRuby::*ire-1a* (344–967aa)::*unc-54* 3'UTR cb-*unc-119*(+)] IV) were generous gifts from the Dillin lab (UC Berkeley).

SJ4005 (zcls4[*hsp-4*p::GFP]), CL2070 (dvIs70[*hsp-16.2*p::GFP]), AM140(rmIs132 [*unc-54*p::Q35::YFP]), AM101 (rmIs110 [F25B3.3p::Q40::YFP]), GRU102 (gnaIs2 [*myo-2*p::YFP + *unc-119*p::Aβ1–42]), TU3401(uIs69 [pCFJ90 (*myo-2*p::mCherry) + *unc-119*p::*sid-1*]), OK814 (*nfyb-1* (*cu13*)), RB1036 (*hyl-1 (ok076)*), GMC101 (dvIs100 [unc-54p::A-beta-1–42::unc-54 3'-UTR + mtl-2p::GFP]), CL2006 (dvIs2 [pCL12(unc-54/human Abeta peptide 1–42 minigene) + rol-6(su1006)]). and N2 wild-type worms and the rest of the strains used were obtained from Caenorhabditis Genetic Center (CGC). All worm strains were grown at 20C using standard methods as described previously [69]. Nematode Growth Media (NGM) agar plates containing OP50 *E. coli* stain were used for normal growth, while HT115 bacteria was used for RNAi feeding. All RNAi clones were obtained from the Ahringer library [70].

### RNAi assays and imaging analysis

Unless stated otherwise, HT115 containing RNAi sequences were cultured overnight in LB containing 100 ug/ml of carbenicillin. Cultures were then seeded onto NGM plates containing 1mM IPTG and stored in a blacked-out box to dry under dark conditions. The RNAi plates must be dried before use and long-term storage to ensure induction of T7 RNA polymerase activity. Worms were then transferred onto RNAi plates during their appropriate time points and grown at 20°C while remaining in the black box away from light. GFP reporter quantification was done by ImageJ. Worms were blindly picked under the LED dissecting microscope, then moved to the M805 Leica Fluorescent microscope to take the image. Briefly, 100 worms were treated per condition, and 6–10 worms were randomly picked under an LED dissecting microscope (blinded); then, worms were transferred to the fluorescent microscope room where they were put into sleep with 2mM levamisole solution. Then the fluorescent images were taken using M805. Then, individual worms were quantified with the ImageJ tools. Three technical repeats and three biological repeats are done for each condition.

## Drug treatment

Worms were synchronized with the bleaching method and grown on OP50 *E. coli* until early adulthood (55 hours post-bleaching). Then, the worms were transferred to RNAi-containing or empty vector control plates. At 72 hours post-bleaching, worms were washed with M9 and treated with tunicamycin (25ug/ml), thapsigargin (2.5uM), or brefeldin A (15ug/ml) for four hours. After the drug treatment, worms were washed with M9 three to five times before transferring them back to fresh RNAi-containing or empty vector control plates. Animals were imaged on day 3 of the adult stage.

## Western blots

Post RNAi treated worms were washed with M9 buffer followed by lysis in RIPA buffer (50 mM Tris-HCl pH 7.5, 150 mM NaCl, 1% NP-40, 0.1% SDS, 2 mM EDTA, and 0.5% sodium deoxycholate) with protease inhibitor cocktail (Roche) at their appropriate time points. The homogenized lysate was then centrifuged at 13,000RPM for 10 minutes at 4°C. The supernatant was then transferred into a new 1.5mL tube to determine protein concentration using BCA assay (Invitrogen). 20–40 ug of protein was loaded per lane in a pre-cast gel for SDS-PAGE (Bio-Rad). Gel was then transferred using the TurboBlot Transfer (Bio-Rad) followed by probing of phosphorylated eIF2α, a-tubulin, VCP and HSP-6. Quantification of bands was further carried out using ImageJ.

## Immunohistochemistry

Worms were fixed by following the previously described Buoin's tube fixation [71] Briefly, 10 plates of worms with RNAi treatment were washed with M9 buffer and distilled water. Pelleted worms were fixed in Bouin's MB fix solution, followed by liquid nitrogen to crack the cuticles. Permeabilized worms were then washed with BTB solution and shaking incubated in BTB solution. BTB solution was replaced with BT solution before incubating with fresh antibody buffer. Antibodies were incubated as described in the antibody buffer. Images were taken using a Nikon A1R confocal laser scanning system equipped with a CIF Plan-Apo 100X objective (NA 1.49, Nikon) and ORCA-Flash 4.0 CMOS camera (Hamamatsu). Primary antibodies: VCP (Abcam), and YFP (Roche). Secondary antibodies: sheep 488 and mouse 647 (Invitrogen)

## Paralysis assay

Adult worms were synchronized using standard bleaching methods as described previously [72]. Worms were grown on standard NGM agar plates containing OP50 at 20C until day 1 adult. On day 1, 100 worms were transferred per RNAi condition onto NGM plates containing RNAi culture and transferred into the 25°C incubator. Worms were transferred onto fresh RNAi plates every other day, and paralysis of the worms was recorded every other day from day 1–5. Following day 5, the paralysis of the worms was recorded every day until all 100 worms were either dead or paralyzed. Worms that died due to external circumstances and of natural causes were censored during data recording to maintain consistency throughout the experiment. Data was recorded and analyzed using Excel and Prism.

## Body bending assay

Am101;Tu3401 worms were grown at 20°C on Nematode Growth Media (NGM) plates spotted with OP50 *E. coli*, then synchronized using the *C. elegans* standard bleaching method as mentioned previously [72]. On day 1 of adulthood, worms were transferred to NGM plates spotted with Empty Vector (EV), *hsp-6*, *hyl-1*, and *hyl-2* RNAi. They were incubated at 20°C (a typical incubation temperature for *C. elegans* growth) in the dark for the rest of the assay to prevent degradation of the IPTG that was inducing the RNAi. Worms were transferred to new RNAi plates every other day and progeny were removed on subsequent days as needed. Motility was measured on day 1 and day 5 of adult lifespan. To measure motility, worms were placed in a drop of M9 media and recorded for 30 seconds under a microscope. The number of body bends each worm performed within the 30 second timespan was manually counted and quantified with ImageJ.

**DAF28::GFP assay**

Daf-28::GFP transgenic worms were synchronized using standardized *C. elegans* bleaching method and eggs were plated on OP50 plates at 20°C to allow for normal development. On day 1 of adulthood, the worms were treated with RNAi targeting *hsp-6*, *xbp-1*, *ire-1* and EV (control). Worms were transferred to new RNAi plates on day 3, followed by imaging of the coelomocytes within the tail region on day 4. GFP intensity was recorded and quantified using ImageJ.

**Statistical analysis**

Statistical analysis was performed as described in the Fig legends. If not specified, paired student's t-test is applied for comparing two groups, and one-way ANOVA with multiple comparisons is applied to for comparing multiple treatments to the control. All assays were repeated with at least three independent experiments. Data distribution was assumed to be normal but was not formally tested.

**Supporting information**

**S1 Fig.** **a)** Inhibition of UPR^ER by *hsp-6* RNAi is regulated through the *dve-1* transcription factor. Animals were treated with tunicamycin as described above for 4 hours on day 1 of adulthood, followed by transfer onto RNAi plates, which targeted specific transcription factors within the mitochondria or cytosolic stress pathway. The animals were imaged at day 3 adult. **b)** Post-development mitochondrial stress through knockdown of different mitochondrial proteins has a different effect on the induction of UPR^ER. Animals were treated with tunicamycin at L4, then were treated with the indicated RNAi or the empty vector control. Animals were imaged at day 3 adult. Graph shows mean+/-SD of four biological repeats, n>=8. **c)** Developmental mitochondrial stress by *cco-1* knockdown also does not affect UPR^ER. UPR^ER (*hsp-4*p::GFP) reporter strains in wild-type (N2) or *nfyb-1 (cu13)* background were treated with control (luciferase) or *cco-1i* RNA from hatch, and GFP expression was measured on day 1 of adulthood with biosorter (left), and the intensity was normalized with the size of the worms (TOF: time of flight). Statistics determined by one-way ANOVA, ns: not significant, * p<0.5, ** p<0.01, Error bar shows mean± s.e.m. **d)** UPR^ER induced by Thapsigargin treatment is not suppressed by MERSR. The graph shows the mean+/-SD of the images of 10-20 animals (three biological replicates). Each RNAi-treated cohort was compared to the Thapsigargin-treated EV control to assess GFP induction. Mock-treated controls are shown on the left, indicating the basal expression of GFP.
(TIFF)

**S2 Fig.** **a)** Tunicamycin treatment does not induce UPR^MT. *hsp-6* reporter animals (SJ4100) were treated with tunica-mycin as described above. The animals were imaged on day 3. The animals were imaged on day 3. **b)** HSP-6 levels following indicated treatment, including RNAi of *hyl-1, hyl-2, and lagr-1* or *hsp-6* reporter animals crossed with *nyfb-1* mutant (*hsp-6*p::GFP;*nyfb*-1). Animals were treated with indicated RNAi on day 1 adult and imaged on day 3. **c)** The *hyl-1* deletion mutant with UPR^ER reporter background. The UPR^ER (*hsp-4*p::GFP) reporter strain (SJ4005) is crossed with *hyl-1 (ok976)* deletion mutant strain to measure UPR^ER activation following tunicamycin treatment. *hyl-1 (ok976)* animals were treated with tunicamycin and RNAi as described above. The graphs show the mean+/-SD of the images of 10–20 animals (three biological replicates). Each RNAi-treated cohort was compared to the tunicamycin-treated EV control to assess GFP induction. Mock-treated controls are shown on the left, indicating the basal expression of GFP. **d)** Western blotting of HSP-6 after double RNAi treatment in Fig 2C. HSP-6 levels were comparable between EV/*hsp-6* double RNAi and *crls-1*/*hsp-6* double RNAi.
(TIFF)

**S3 Fig.** **a)** *pek-1* mediates the suppression of UPR^ER. Animals were treated with tunicamycin and the indicated RNAi as previously described (Fig 1). Graph shows mean+/-SD of three biological repeats, n>=6. **b)** (left) Western blot of *daf-28*p::GFP and DAF-28::GFP shows that *hsp-6*, *ire-1*, *xbp-1* or *cco-1* RNAi treatment does not change *daf-28*

transcriptional and translational expression levels, suggesting that the reduction of *daf-28* exhibited in the coelomocytes in Fig 3F is most likely the result of a decrease in ER secretory function. (right) DAF-28::GFP secretion to coelomocytes were unchanged with UPRMT induced by *cco-1* RNAi. The graph shows mean +/-SD of GFP intensity normalized to empty vector control, n>=10 with three biological repeats. **c)** UPR$^{ER}$ induced by Brefeldin A treatment is also suppressed by MERSR. The graph shows the mean +/-SD of the images of 10–20 animals (three biological replicates). Each RNAi-treated cohort was compared to the Brefeldin A-treated EV control to assess GFP induction. Mock-treated controls are shown on the left, indicating the basal expression of GFP. **d)** Proteotoxicity models expressing Abeta (1–42) or polygluta-mine (Q35) within body wall muscles. Individual RNAi of UPR$^{ER}$ components followed by paralysis assay (left) and motility assay (right).
(TIFF)

**S4 Fig.** **a) (**left) Lifespan of the animals expressing polyQ35::YFP in their body wall muscle (AM140) with constitutively active UPR$^{ER}$ in the whole-tissue by expressing *xbp-1s* (*myo-*3p::polyQ35::YFP; sur-*5p::xbp-1s,myo-2*p::tdTomato). Log-rank * $p < 0.05$ for EV vs *hyl-2*. (right) Motility was determined using a body bending assay that measures the number of body bending per 30 seconds. The relative motility was plotted by normalizing with empty vector control, and the rate was compared to that of the empty vector control (Three biological repeats with n > 10, mean +/- SD). Statistics determined by t-test comparing each condition to EV control, * $p < 0.5$, ** $p < 0.01$. **b)** Paralysis assay of GMC101 animals that express Aβ (1–42) in their body wall muscle. The assay was performed at 25°C as previously described [73]. *hsp-6* was knocked down from day 1 of adulthood. **c)** (left) Lifespan of wild-type N2 worms treated with *hsp-6* RNAi. Animals were trans-ferred to RNAi-containing plates on day 1 adult. $P < 0.001$. (right) Lifespan *nfyb-1(cu13)* mutant animals following *hsp-6* knockdown from day 1 of adulthood. *hsp-6* RNAi moderately extended the lifespan (Log-rank ****$p < 0.0001$) of wild-type animals, whereas it did not significantly affect the lifespan of *nfyb-1* mutant animals.
(TIFF)

**S5 Fig. A working model of the proposed mechanism of MERSR.** Our findings suggest that the mitochondrial stress response (MCSR), triggered by HSP-6 knockdown, modulates UPR$^{ER}$ signaling pathways through alterations in ceramide and cardiolipin levels. As misfolded proteins build up in the ER, the mitochondrial stress response curtails IRE-1 activation and turns-off the expression XBP1 target genes. Simultaneously, it promotes the PERK-dependent eIF2α phosphoryla-tion, leading to a reduction in global protein translation. Consequently, as a newly discovered branch of MCSR, MERSR increases the ER stress threshold and the ER's protein processing capacity, enhancing overall cellular proteostasis. Cre-ated in BioRender. Kim, H. (2025) https://biorender.com/n8x93hs.
(TIFF)

**S1 Table. Raw data of imaging analyses Raw data of the imaging analyses, including biological repeats, are listed in the table.** Each figure is listed in the separately labeled tab.
(XLSX)

## Acknowledgments

We thank the Caenorhabditis Genetic Center and Shohei Mitani of the National BioResource Project for providing strains. We thank Dr. Daniel Garsin from University of Texas Health Science Center at Houston for sharing RNAi clones and other reagents. We thank Dr. Andrew Dillin and lab members for sharing the strains with us. We thank Jina Park, an English major Rice University student, for proofreading the manuscript. We thank Yelena Robinson, a Rice University student, and Tuckger Engels, a Medical student at the University of Texas Health Science Center at Houston (McGovern Medical School), for tech-nical support. Fluorescence microscopy was performed at the Nikon Center of Excellence - Center for Advanced Microscopy, Department of Integrative Biology & Pharmacology at McGovern Medical School, UTHealth Houston.

## Author contributions

**Conceptualization:** Hyun-Eui Kim.

**Data curation:** Jeson J. Li, Nan Xin, Bo G Kim, Travis I. Moore, Hyun-Eui Kim.

**Formal analysis:** Jeson J. Li, Nan Xin, Chunxia Yang, Bo G. Kim, Larissa A. Tavizon, Ruth Hong, Jina Park, Travis I Moore, Rebecca George Tharyan.

**Funding acquisition:** Hyun-Eui Kim.

**Investigation:** Jeson J. Li, Nan Xin, Chunxia Yang, Bo G. Kim, Larissa A. Tavizon, Ruth Hong, Jina Park, Rebecca George Tharyan, Hyun-Eui Kim.

**Methodology:** Nan Xin, Hyun-Eui Kim.

**Resources:** Adam Antebi, Hyun-Eui Kim.

**Supervision:** Adam Antebi, Hyun-Eui Kim.

**Validation:** Jeson J. Li, Nan Xin, Bo G. Kim.

**Visualization:** Travis I. Moore.

**Writing – original draft:** Jeson J. Li, Hyun-Eui Kim.

**Writing – review & editing:** Jeson J. Li, Nan Xin, Rebecca George Tharyan, Adam Antebi, Hyun-Eui Kim.

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
