## [Decision Letter · Decision Letter 0]

22 Oct 2024

Dear Dr Kim,

Thank you very much for submitting your Research Article entitled 'Unveiling the Intercompartmental Signaling Axis: Mitochondrial to ER Stress Response (MERSR) and its Impact on Proteostasis' to PLOS Genetics.

The manuscript was fully evaluated at the editorial level and by independent peer reviewers. The reviewers appreciated the attention to an important problem, but raised some substantial concerns about the current manuscript. Based on the reviews, we will not be able to accept this version of the manuscript, but we would be willing to review a much-revised version. We cannot, of course, promise publication at that time.

If you decide to revise the manuscript for further consideration at PLOS Genetics, please aim to resubmit within the next 60 days, unless it will take extra time to address the concerns of the reviewers, in which case we would appreciate an expected resubmission date by email to plosgenetics@plos.org.

If present, accompanying reviewer attachments are included with this email; please notify the journal office if any appear to be missing. They will also be available for download from the link below. You can use this link to log into the system when you are ready to submit a revised version, having first consulted our Submission Checklist .

PLOS has incorporated Similarity Check , powered by iThenticate, into its journal-wide submission system in order to screen submitted content for originality before publication. Each PLOS journal undertakes screening on a proportion of submitted articles. You will be contacted if needed following the screening process.

To resubmit, log into your Editorial Manager account and select the option 'Revise Submission' in the 'Submissions Needing Revision' folder.

We are sorry that we cannot be more positive about your manuscript at this stage. Please do not hesitate to contact us if you have any concerns or questions.

Yours sincerely,

Monica P. Colaiácovo

Section Editor

PLOS Genetics

Reviewer's Responses to Questions

**Comments to the Authors:**

Reviewer #1: This is an interesting study. In previous work, the authors discovered that mitochondria can communicate their proteostasis status to the cytosol, activating cytosolic stress responses that enhance proteostasis in the cytosol. In this manuscript, the authors provide evidence demonstrating that perturbations in mitochondrial proteostasis also influence ER stress responses, a cross-compartmental pathway that the authors call MERSR. Moreover, they identified the ER pathways modulated by alterations in mitochondrial proteostasis (i.e., inhibition of the IRE1 branch and activation of the PERK-eIF2α pathway). They also found that MERSR activation in various disease models ameliorates disease-related protein aggregation and subsequent pathological changes. Together, this study has significant implications for proteostasis and neurodegenerative diseases involving protein aggregation. Nevertheless, I have several comments that I hope the authors will address.

- In the first paragraph of the introduction, the authors briefly mention molecular chaperones and the suppression of global protein synthesis as proteostasis mechanisms. However, they should also include proteolytic systems (especially because ERAD seems to be relevant for the effects reported).

- In the introduction: ‘which compensates for the increase in proteomic stress…’ I’m not sure if ‘proteomic stress’ is correct. This is also used in many other instances throughout the text.

- To further strengthen their conclusions from Figure 2, can the authors test if mtHSP70 RNAi does not further decrease UPRER activation in HYL-1, or HYL-2 or LAGR-1 mutants (any of these genes would be enough, if there available mutant strains for any of them).

- Can the authors test knockdown levels in Figure 2c to confirm that the knockdown levels for hsp-6 are similar in hsp-6/EV and hsp-6/crls-1?

- In the results sections, many text appear to be more appropriate for the Discussion section. For example: “This suggests that the suppression of UPRER occurs upstream of xbp-1, possibly at a membrane level where IRE1 resides. One possibility is that mtHSP70 knockdown leads to an alteration in membrane fluidity, compromising the dimerization of the UPRER sensor IRE1. This is further corroborated by recent findings that the knockdown of ER-resident proteins alters the ER membrane lipid content, increasing mitochondria and ER contact through increasing membrane order 26. Furthermore, alterations in sphingolipid, including ceramide content within the ER membrane, have been shown to induce UPRER in the absence of aberrant proteostasis 27, providing further evidence that membrane lipid content can directly influence the UPRER signaling pathways”.

- This paragraph in the results section is too long and contains information not directly relevant for the results sections (e.g. discussion about earliest AB models): “Dysfunctional ER stress is a common phenotype in many neurodegenerative diseases, including Alzheimer’s, Huntington’s, and Parkinson’s Disease 35. Early notions of ER stress denoted its function as deleterious 36; however, recent studies have suggested that the response of UPRER in neurodegenerative diseases is highly dynamic 37. Mild UPRER is upregulated during early stages of AD in order to resolve amyloid-�� (A�) aggregate deposition, however, chronic A� burden as a result of advanced AD and constitutive UPRER activation triggers downstream apoptotic pathways, leading to cell death 38. A� metabolism is a constant cycle, starting from the synthesis of Amyloid Precursor Protein (APP) in the ER, followed by the export to the Golgi and ultimately to the plasma membrane where it is cleaved by beta-secretase and gamma-secretase to form mature A� 39. While an ortholog of APP containing beta-secretase sites is absent in C. elegans, several disease models expressing the disease-causing human A� (1-42) peptide have been generated. One of the earliest models of A��worms was thought to have been able to generate A� (1-42) peptides 40, but further research established that the final A� product was a truncated version (3-42) due to mis-cleavage of a synthetic signal peptide 41. However, by modifying the artificial signal peptide on the A� plasmid, worm strains that produce A� (1-42) peptide have now been generated 42.

- This sentence appears before the results are explained: “Further evidence suggests that VCP is highly correlated with densely packed polyglutamine aggregates, and upon MERSR activation, a portion of VCP becomes highly associated with loosely packed polyglutamine aggregates”. In the same paragraph, some sentences are more speculative and could fit better into the discussion section. I would suggest to reorganize/rewrite the whole paragraph for clarity.

- In Figure S2C, the authors focused on lifespan assays. As a more direct test of physiological consequences in polyQ-expressing worms, the authors could perform motility assays.

Reviewer #2: In this manuscript the authors expand on their previous findings of a Mitochondrial to Cytosolic Stress Response, and describe a new mitochondrial to ER crosstalk of relevance for C. elegans health. They show that upon ER stress, hsp-6 knockdown impact on both UPRER signaling branches through alteration of lipids (cardiolipin and ceramide) homeostasis: on the one hand hsp-6 RNAi suppress IRE-1 activation of XBP1-regualted genes and on the other it induces PERK-dependent eIF2α phosphorylation, in turn reducing global protein translation. Suppression of these ER regulated signaling axes ultimately by hsp-6 knock down positively impact cellular proteostasis and nematode healthspan in models of proteinopathies. The findings may be thus of interest for a broad audience.

The study is interesting and shed light on a novel intercompartmental signaling axis of relevance for animal’s health. Nonetheless, additional experiments are necessary to substantiate authors’ conclusions before considering the work for publication.

1) What is the effect of hsp-6 RNAi on hsp-4 expression in basal (untreated conditions). Do other ER stressors such as thapsigargin or brefeldin A induce hsp-4 expression in a hsp-6-dependent manner? Also, does overexpression of hsp-6 per se turn on ER stress response?

2) On the other hand, and perhaps more importantly, the first results chapter the author conclude that ER stress is suppressed through the activation of MERSR. To support this conclusion, they should show if ER stress actually activate a MERSR. Namely, what is the effect of ER stressors on hsp-6 expression and more generally on mitochondrial function? Also, mitochondria stress-induced UPR promotes lifespan extension. Does ER-stress also promote lifespan or suppress it? This is also in light of the findings on proteotoxicity.

3) On the same line, at the end of the first results chapter (Figure S1b) the authors show that mrps-5, spg-7 or cco-1 knockdown was unable to reduce UPRER signaling during ER stress. Knockdown of mitochondrial proteins however classically increases hsp-6 expression, which would be expected (opposite to the author speculate) to in fact stimulate hsp-4 expression rather than suppressing it. Why is this not the case?

4) In line with previous comment, does alteration of lipids homeostasis (eg knock down of ceramide synthases) affects expression of hsp-6 in basal conditions or upon ER stress? And viceversa, are the two ER stress modulated branches affected by other conditions which impact on hsp-6 expression?

5) Does mitochondrial stress induced mtUPR (or overexpression of hsp-6) increase the expression of GFP within the coelomocytes or of DAF-28 expression?

6) Previous work has clearly demonstrated that induction of the mtUPR has beneficial effects against Abeta induced proteotoxicity. Nature. 2017 Dec 14;552(7684):187-193. doi: 10.1038/nature25143. How do the authors reconcile their findings with previously published work? Does hsp-6 knockdown impact on wild-type animals’ lifespan in basal conditions or upon ER-induced stressors?

7) Does amelioration of the proteotoxicity also rely on different components of the ER-stress response branches ie suppress IRE-1 and activation of XBP1-regualted genes and PERK-dependent eIF2α pathway?

Minor comments

- In the introduction the authors refer to own previous work indicating intercompartmental crosstalk. They should cite the related studies.

- It is not clear why the authors in the text write mtHSP70 while in fact they are working and referring to it in the figure to C. elegans hsp-6 RNAi. Should keep it consist for simplicity.

- The authors report (Figure S1a) that dev-1RNAi also suppresses ER stress response induction. Why this is also not the case for atfs-1?

- In figure 4 the authors use opposite colors for EV and hsp-6 legends, which is somewhat confusing.

- In the last results’ chapter, the authors use a strain which they claim to be neuronal RNAi-sensitive. Yet this has been reportedly shown to display RNAi sensitivity also in other tissues. Results should be therefore discussed accordingly.

- MERC and ceramides are involved in apoptosis. It would be intersting to know whether any of the effects modulated by MERSR are mediated by apoptotic-regulatory genes?

**Have all data underlying the figures and results presented in the manuscript been provided?**

Reviewer #1: Yes

Reviewer #2: Yes

PLOS authors have the option to publish the peer review history of their article (what does this mean? ). If published, this will include your full peer review and any attached files.

**Do you want your identity to be public for this peer review?** For information about this choice, including consent withdrawal, please see our Privacy Policy .

Reviewer #1: **Yes: ** David Vilchez

Reviewer #2: No

---

## [Decision Letter · Decision Letter 1]

24 Apr 2025

Dear Dr Kim,

We are pleased to inform you that your manuscript entitled "Unveiling the Intercompartmental Signaling Axis: Mitochondrial to ER Stress Response

(MERSR) and its Impact on Proteostasis" has been editorially accepted for publication in PLOS Genetics. Congratulations!

Yours sincerely,

Monica P. Colaiácovo

Section Editor

PLOS Genetics

Aimée Dudley

Editor-in-Chief

PLOS Genetics

Anne Goriely

Editor-in-Chief

PLOS Genetics

Comments from the reviewers (if applicable):

Reviewer's Responses to Questions

**Comments to the Authors:**

Reviewer #1: The authors have addressed all my comments and I support publication of the manuscript

**Have all data underlying the figures and results presented in the manuscript been provided?**

Reviewer #1: Yes

PLOS authors have the option to publish the peer review history of their article (what does this mean? ). If published, this will include your full peer review and any attached files.

**Do you want your identity to be public for this peer review?** For information about this choice, including consent withdrawal, please see our Privacy Policy .

Reviewer #1: **Yes: ** David Vilchez

**Data Deposition**

http://datadryad.org/submit?journalID=pgenetics&manu=PGENETICS-D-24-01069R1

**Press Queries**

---

## [Editor Report · Acceptance letter]

PGENETICS-D-24-01069R1

Unveiling the Intercompartmental Signaling Axis: Mitochondrial to ER Stress Response

(MERSR) and its Impact on Proteostasis

Dear Dr Kim,

We are pleased to inform you that your manuscript entitled "Unveiling the Intercompartmental Signaling Axis: Mitochondrial to ER Stress Response

(MERSR) and its Impact on Proteostasis" has been formally accepted for publication in PLOS Genetics! Your manuscript is now with our production department and you will be notified of the publication date in due course.

With kind regards,

Anita Estes

PLOS Genetics

On behalf of:
